# *SELENOP* rs3877899 Variant Affects the Risk of Developing Advanced Stages of Retinopathy of Prematurity (ROP)

**DOI:** 10.3390/ijms24087570

**Published:** 2023-04-20

**Authors:** Ewa Strauss, Danuta Januszkiewicz-Lewandowska, Alicja Sobaniec, Anna Gotz-Więckowska

**Affiliations:** 1Institute of Human Genetics, Polish Academy of Sciences, Strzeszynska 32, 60-479 Poznan, Poland; 2Department of Pediatric Oncology, Hematology and Transplantology, Poznan University of Medical Sciences, Szpitalna 27/33, 60-572 Poznan, Poland; danuta.januszkiewicz@ump.edu.pl; 3Department of Medical Diagnostics, Dobra Street 38a, 60-595 Poznan, Poland; 4Department of Neonatology, Poznan University of Medical Sciences, Polna 33, 60-535 Poznan, Poland; 5Department of Ophthalmology, Poznan University of Medical Sciences, Szamarzewskiego 84, 60-569 Poznan, Poland; agotzwieckowska@ump.edu.pl

**Keywords:** preterm newborn, retinopathy of prematurity, ROP, *SELENOP*, selenoprotein P

## Abstract

The significance of selenoproteins for the incidence of prematurity and oxidative-damage-related diseases in premature newborns is poorly understood. The latter are at risk for ROP as well as BPD, IVH, PDA, RDS, and NEC, which is particularly high for newborns with extremely low gestational age (ELGA) and extremely low birth weight (ELBW). This study evaluates the hypothesis that variation in the selenoprotein-encoding genes *SELENOP*, *SELENOS*, and *GPX4* affects the risk of ROP and other comorbidities. The study included infants born ≤ 32 GA, matched for onset and progression of ROP into three groups: no ROP, spontaneously remitting ROP, and ROP requiring treatment. SNPs were determined with predesigned TaqMan SNP genotyping assays. We found the association of the *SELENOP* rs3877899A allele with ELGA (defined as <28 GA), ROP requiring treatment, and ROP not responsive to treatment. The number of RBC transfusions, ELGA, surfactant treatment, and coexistence of the rs3877899A allele with ELGA were independent predictors of ROP onset and progression, accounting for 43.1% of the risk variation. In conclusion, the *SELENOP* rs3877899A allele associated with reduced selenium bioavailability may contribute to the risk of ROP and visual impairment in extremely preterm infants.

## 1. Introduction

Prematurity is one of the major health-threatening problems in neonates [1]. Preterm babies are highly susceptible to oxidative stress due to an imbalance between the oxidant and antioxidant systems. The generation of free radicals induces oxidative damage to multiple body organs and systems and results in the development of typical premature infant diseases, including bronchopulmonary dysplasia (BPD), intraventricular hemorrhage (IVH), patent ductus arteriosus (PDA), respiratory distress syndrome (RDS), necrotizing enterocolitis (NEC), diffuse white matter injury (DWMI), or retinopathy of prematurity (ROP). The risk is particularly high for newborns with extremely low gestational age (ELGA) and extremely low birth weight (ELBW) [2,3]. 

Selenium (Se) and selenoproteins perform an essential function in fetal development and health by regulating antioxidant and inflammatory processes [4,5,6,7]. Se’s nutritional status of the fetus and newborn depends on the mother’s antioxidant status. Uptake of this micronutrient in the fetus occurs via placental transfer, mainly during the 3rd trimester of pregnancy. Se is obtained through diet, and its plasma concentrations vary by country depending on its content in soil and Se-containing supplement intake [8]. Concentrations of Se observed in European adults and infants are relatively low, which may contribute to the relatively high impact of Se deficiency on the occurrence of oxidative stress-related diseases in European populations. 

There is evidence that cord blood Se levels positively correlate with gestational age (GA), birth weight (BW), and Apgar score at 5 min, confirming the protective role of Se against the occurrence of preterm delivery [9]. Consequently, lower Se levels in preterm infants than in term-born ones are observed. This deficiency can be further exacerbated in the postnatal period due to prolonged parenteral nutrition and poor intestinal absorption. Considering that Se concentrations increase in healthy breastfed infants after birth, the Se deficiency found in preterm infants must be clinically important [10]. The bioavailability of Se in the body and redox homeostasis are partly determined by selenoprotein activity, which is under genetic control [5,11]. The specific role of selenoproteins in development is indicated by observations of a phenotype of severe gene mutations in the selenoprotein biosynthesis pathway that result in global developmental delay [12]. 

The relationship between variants in genes encoding selenoproteins and the risk of developing ROP and other complications of prematurity has not yet been investigated. In this study, we evaluated the association of four SNPs in the genes encoding selenoprotein P (SelP, *SELENOP* gene), selenoprotein S (SelS, *SELENOS* gene), and glutathione peroxidase 4 (GPX4, *GPX4* gene) with the development of ROP and its clinical course. The associations and interactions between these variants and the presence of other comorbidities (BPD, IVH, PDA, RDS, and NEC) and extremely low gestational age and birth weight were also investigated. 

## 2. Results

### 2.1. Clinical Risk Factors of ROP Development and Unsuccessful Treatment

Population characteristics are shown in Figure 1, while risk factors for developing ROP and treatment failure are shown in Table 1. In the study group of 190 children, as many as 173 (91.0%) completed the entire neonatal and ocular follow-up (Figure 1). The remaining 17 children (9.0%) were excluded from the study. Among them, 7 (3.7%) died, and another 10 (5.3%) were lost in follow-up. 

Ultimately, 121 children with ROP were included in the group. Among them, 55 (31.8%) were diagnosed with ROP that regressed spontaneously, and 66 (38.2% of the group) had ROP requiring treatment. In the latter group, LP was performed as the first line of treatment in 46 infants (69.7%); 19 infants (28.8%) received IVR injections; and 1 infant (1.5%) was treated simultaneously with LP and IVR. Treatment failure was observed in 19 cases of ROP (11.0%). In the studied group, ROP was diagnosed on average at 52.9 ± 12.4 days of life (7.6 weeks), at a mean post-menstrual age (PMA) of 34.4 ± 2.0 weeks.

The presence of ROP and its progression to advanced stages was negatively correlated with GA (β = 0.591; *p* < 0.0001), BW (β = 0.509; *p* < 0.0001), and Apgar scores at 1 and 5 min (β = 0.323; *p* < 0.0001 and β = 0.267; *p* < 0.001), while being positively correlated with the number of blood transfusions (β = 0.554; *p* < 0.0001), duration of mechanical ventilation (β = 0.548; *p* < 0.0001), the occurrence of other factors related to respiratory failure (treatment with surfactant and resuscitation), and the presence of comorbidities: BPD (β = 0.477; *p* < 0.0001), IVH (β = 0.364; *p* = 0.0001), sepsis (β = 0.296; *p* < 0.001), RDS (β = 0.262; *p* = 0.001), and NEC (β = 0.234, *p* = 0.002, Table 1). There were no significant correlations between risk factors present at birth (i.e., fetal bladder rupture and delivery by cesarean section) and the development and progression of ROP. Among the neonates with ROP requiring treatment, the only factor significantly associated with failure of laser therapy was multiple blood transfusions (8.0 ± 3.6 vs. 4.5 ± 2.0 *p* = 0.026); other factors showing an association at the level of a statistical trend were low BW (*p* = 0.191) and the coexistence of IVH (*p* = 0.142). All these children required resuscitation and mechanical ventilation. Details of the relationship between age and the occurrence of ROP are shown in Figure 2.

### 2.2. Polymorphisms in Selenoprotein Genes and the Incidence, Severity, and Treatment of ROP

The genotype distributions of the studied SNPs showed no deviations from the Hardy–Weinberg equilibrium. The studied *SELENOP* SNPs were found to be in significant linkage disequilibrium (r = 0.348; *p* < 0.0001). An association was found between the *SELENOP* SNPs and the occurrence of advanced ROP and/or treatment failure (Figure 3a and Appendix A). The frequency of the *SELENOP* rs3877899A allele in premature infants without ROP was 0.202; in those with spontaneous regression of ROP, it was 0.255; in those with proliferative ROP, it was 0.318; and in cases with unsuccessful treatment, it was as high as 0.395. The corresponding frequencies of rs3877899AA homozygotes were: 1.9%, 9.1%, 12.1%, and 21.1%. Among the ROP cases requiring treatment, 50% of rs3877899AA homozygotes experienced treatment failure.

Statistically significant associations were observed for the rs3877899A allele and the rs3877899AA homozygotes and the occurrence of ROP requiring treatment (OR = 1.8, *p* = 0.045; OR = 7.0, *p* = 0.038, respectively) or ROP with treatment failure (OR = 2.6, *p* = 0.019; OR = 13.6, *p* = 0.005, respectively, in relation to the clinical course without ROP), Figure 3b. On the other hand, the *SELENOP* rs3877899A allele was associated with a lower risk of ROP treatment failure. However, this effect may be partly attributed to linkage disequilibrium between the studied SNPs. The frequency of the *SELENOP* rs7579A allele was 0.324 in preterm infants without ROP and 0.132 in infants with unsuccessful ROP treatment (OR = 0.32; *p* = 0.023). A protective effect was observed in the dominant model (OR = 0.20, *p* = 0.008). The other studied SNPs in the *SELENOS* and *GPX4* genes were not associated with ROP occurrence, progression, or treatment effectiveness.

### 2.3. Polymorphisms in Selenoprotein Genes and the Incidence of Other Complications of Prematurity

RDS, IVH, BPD, PDA, NEC, LBW, ELGA, and ELBW were studied. An association between *SELENOP* rs3877899 and ELGA was observed (Figure 3c; Appendix A). More than 52% of infants born before 28 weeks of gestation carried the rs3877899A allele, compared with 38% of infants born after at least 28 weeks of gestation (*p* = 0.02). Carriers of this allele had a 1.9-fold increased risk of being born with ELGA (*p* = 0.04). No other associations were found between the selenoprotein genotype and the occurrence of other complications of prematurity.

### 2.4. Gene-Environment Interaction between the SELENOP Genotype and ELGA

A gene-environment interaction between the *SELENOP* rs3877899 genotype and ELGA affected the risk of developing ROP and its progression to advanced stages. The co-occurrence of the rs3877899GA+AA genotype with ELGA was observed in 3.8% of cases without ROP, in 16.4% of cases with spontaneously regressive ROP, and in 45.5% of cases with ROP requiring treatment (Table 2; Figure 3d). Compared to children with the rs3877899GG genotype who were born ≥28 GA, children with both risk factors (the rs3877899GA+AA genotype and ELGA) had a 35-fold increased risk of proliferative ROP (*p* < 0.0001). The Rothman synergy index in both cases indicated a multiplicative interaction. The co-occurrence of the studied risk factors was associated with a 1.6-fold and 3.1-fold greater impact with respect to the sum of their separate influences. There was no effect of the coexistence of these risk factors on ROP treatment. Both risk factors were present in a similar proportion of successfully treated children (n = 21; 45%) and in the subgroup with treatment failure (n = 9; 47.4%).

Similar effects were observed for the co-occurrence of the *SELENOP* rs3877899GA+AA genotype with ELBW, but they were less significant, indicating a secondary effect in relation to age. The coexistence of the rs3877899GA+AA genotype and ELBW was observed in 5.7% of children without ROP, 16.4% of children with spontaneously regressive ROP, and 36.4% of children with ROP requiring treatment. These frequencies correspond to a 5-fold (95% CI: 1.2–22.2; *p* = 0.024) and 20-fold (95% CI: 5.1–81.6); *p* < 0.0001) increase in the risk of developing spontaneously regressive ROP or ROP requiring treatment. Only for ROP requiring treatment did the observed relationships indicate a multiplicative interaction (S = 1.5).

### 2.5. Independent Predictors of the Occurrence and Progression of ROP

The results of the multivariate analysis for the occurrence and progression of ROP are presented in Table 3. 

Factors showing significant association in univariate analysis were selected for further evaluation; they included ELGA, ELBW, Apgar score at 1 min, number of RCB transfusions, surfactant treatment, resuscitation, mechanical ventilation, duration of mechanical ventilation, presence of NEC, BPD, IVH, RDS, sepsis, or the *SELENOS* rs3877899 genotype (A allele dose), and co-occurrence of the *SELENOS* rs3877899GA+AA genotype with ELGA. All factors examined were included in model 2, whereas model 1 did not include the co-occurrence of the rs3877899GA+AA genotype with ELGA. Regression analysis demonstrated that the most important risk factors for the onset and progression of ROP were the number of RCB transfusions, ELGA, and surfactant treatment, which jointly accounted for 41.3% of the risk variation (30.0%, 8.5%, and 2.1%, respectively). When considering the combined effect of genotype and age, this increased risk variability by 1.8%.

## 3. Discussion

ROP is a disorder resulting from abnormal retinal vascularization and is a leading cause of visual impairment and blindness in children [13]. The overall incidence of any stage of ROP ranges from 15% to 68%, depending on the studied population. It is diagnosed on average at 8 wk. of life (34–35 PMA). Lower GA and BW are major risk factors for the development of the disorder. Among ELBW infants, the observed percentage of any stage ROP reaches 80%, and the percentage of ROP cases requiring treatment reaches 24%; in infants with GA ≤ 25 weeks, these rates amount to 100% and 67%, respectively [14,15]. The frequency of blindness in children attributable to ROP ranges from 10% to 37.4% worldwide [16].

The mechanism through which retinal lesions develop in premature infants is well documented. It is linked to a lack of effective regulation of the retinal and choroidal circulation, which promotes retinal vasoconstriction, endothelial cell degeneration, inflammation, and uncontrolled neovascularization. The pathogenesis of these changes is multifactorial and includes damage induced by oxidative stress and resulting from neonatal immaturity and related factors. Among them, the presence of RDS, prolonged oxygen therapy, and multiple blood transfusions are the most important [15]. The results of this study are consistent with previous clinical observations: low GA was the strongest predictor of ROP presence and progression to advanced stages, followed by a high number of blood transfusions and prolonged mechanical ventilation. Numerous blood transfusions were also a risk factor for ROP treatment failure. Among the comorbidities, the strongest correlation with ROP was shown by BPD and IVH, which suggests a common etiopathogenesis. The number of RBC transfusions, ELGA, and surfactant treatment proved to be independent predictors of ROP onset and progression, accounting for 41.3% of the risk variability in the studied population.

The role of Se and genetic variation in enzymes involved in maintaining optimal function of the antioxidant system [6] and susceptibility to complications induced by oxidative stress [17,18,19] in premature infants was previously demonstrated, but the impact of selenoproteins has yet to be investigated. Therefore, this study focused on the role of genetic variants for selenoproteins that act as antioxidants in plasma (SeP) and in the intracellular space, including the endoplasmic reticulum (SelS), cytoplasm (GPX4), and mitochondria (GPX4). The studied variants affect protein function or concentration in the body and have been previously linked to cancer and cardiovascular disease [20,21,22,23,24,25]. Among them, only SNPs in *SELENOP* were found to be associated with ROP. SeP is a multifunctional protein that not only acts as an antioxidant but also stores and transports Se. Unlike other selenoproteins, which contain only a single selenocysteine incorporated into the N-terminal domain, it possesses multiple selenocysteine residues (up to 9) in the C-terminal domain [26].

A summary of the univariate analysis results is presented in Figure 2. The rarer allele of the rs3877899 SNP (variant A) was found to be a risk factor for proliferative, more advanced ROP. In contrast, the rarer allele of the rs7579 SNP (variant A) played a protective role; however, this effect was probably a consequence of linkage disequilibrium between the studied SNPs. The rs3877899A allele was also associated with ELGA in the studied children, and both factors (age and the rs3877899 genotype) interacted to further increase the risk of ROP. In univariate analysis, the coexistence of the rs3877899GA+AA and ELGA genotypes further amplified the risk of spontaneously regressive ROP and ROP requiring treatment (by factors of 1.6 and 3.1, respectively). After adjusting for the influence of other risk factors, the combined effect of the risk genotype and ELGA explained 1.8% of the variation in ROP incidence and progression. This impact was similar to the influence of surfactant use, which amounted to 2.1%. It can be assumed that rs3877899A coexisting with ELGA is also a significant risk factor for ROP treatment failure. Treatment was required primarily for children of low birth age, and the children whose treatment proved unsuccessful were characterized by an extremely high frequency of the rs3877899A allele (0.395 vs. 0.287 in cases treated successfully) and its homozygotes (21.1% vs. 8.5% in patients treated successfully; significance level for both *p* = 0.2). The frequency of the rs3877899A allele in the subgroup of children with ROP treatment failure was significantly higher than in the subgroup without ROP (0.395 vs. 0.202; *p* = 0.019), which featured only a small percentage of children with ELGA.

The observed effects of SNPs were consistent with expectations based on previously described allelic functions. The studied SNPs were found to alter the synthesis ratio of the two isoforms of SeP, which differ in size and Se content. The rs3877899A and rs7579G alleles favor the production of a 60-kDa Se-rich isoform with 10 Sec residues, while the rs3877899A and rs7579G alleles favor the production of a 50-kDa Se-poor isoform containing only one Sec residue [27]. The rs3877899 variant is also involved in the regulation of protein stability and cellular protein uptake. These SNPs were found to affect selenium’s bioavailability for the synthesis of all other selenoproteins by influencing the body’s Se status and transport, as well as the effectiveness of supplementation [28,29,30]. Our findings regarding the association of the rs3877899A allele with ELGA are also in line with previous clinical studies in the Italian population, which demonstrated the association of SNPs in antioxidant enzymes with low GA and/or BW (for superoxide dismutase 2) as well as complications of prematurity, including ROP (for superoxide dismutase 1, 3, and catalase) [7,31]. Both the latter results and our own findings suggest that the development of the antioxidant system during fetal life is significantly involved in the maintenance of pregnancy through uteroplacental-fetal interactions. After birth, this system can be crucial for proper retinal function and development. Possible mechanisms are shown in Figure 4.

The importance of selenoproteins for development is illustrated by the embryonic lethal phenotype of knockout mice for the genes of the selenoprotein biosynthesis pathway. In humans, carriers of mutations in these genes (including *SECISBP2*, *TRU-TCA1-1,* and *SEPSECS*) display the severe multisystem phenotypes of global developmental delay with features of myopathy and cerebellar and optic atrophy [12].

The examined *SELENOS* and *GPX4* variants were not associated with complications of prematurity, ELGA, or ELBW. The effects of these SNPs are likely too weak to be demonstrated in a small group. On the other hand, the adverse effect of the rs3877899A allele is associated with decreased production of other selenoproteins and may exacerbate the effect of carrying other unfavorable genetic variants. It may contribute to increased production of pro-inflammatory cytokines [4,21,31], lipid peroxides [32], and antioxidant barrier depression [20,33,34], related to the genetic deficiency in SeP and GPX4.

The specific role of Se and selenoproteins in the development of diseases associated with prematurity is still poorly understood. The literature data indicate that Se is an important contributor to ROP [10], pregnancy outcomes [35], the risk of intrauterine growth retardation and preterm birth [36], as well as the functioning of the immune and antioxidant systems in preterm infants [37]. Recent data confirm the significant role of oxidative stress in both the incidence and severity of ROP [38].

Given the lack of guidelines for standardized Se monitoring methods and the absence of Se availability for supplementation, it is reasonable to conclude that Se deficiency in extremely preterm infants is probably common and underdiagnosed [6]. ELGA and *SELENOP* genotypes may be important factors in identifying preterm infants at high risk of developing advanced ROP. Whether Se supplementation will prevent advanced ROP as well as avoid morbidity and mortality in preterm infants remains an open question [35]. Randomized clinical trials have shown that postnatal Se supplementation in very low birth weight infants can reduce one or more episodes of sepsis. However, such supplementation has not improved neonatal outcomes in terms of ROP, survival, or lung diseases [39]. These outcomes may be due to insufficient dosages of supplementation, a lack of maternal supplementation during pregnancy, or the modifying effects of the *SELENOP* genotype. The influence of genotype was previously demonstrated in the UK population, where pregnant women who carry the rs3877899A allele could better maintain their Se status during pregnancy and were more responsive to Se supplementation than carriers of the rs3877899G allele [29]. The impact of the rs3877899A allele on Se supplementation was also observed in women from the general population [28]. These results are promising for the potential use of protective supplementation during pregnancy. However, developing a personalized therapy approach presents a challenge as it would require clinical trials involving several thousand pregnant women. The supplementation of another micronutrient, zinc, in preterm infants was significantly associated with reduced mortality but not comorbidities such as BPD, ROP, bacterial sepsis, or NEC [40]. A limitation of the aforementioned studies in the infant cohorts was the low prevalence of the diseases studied. For example, in one study on ROP involving 193 premature infants, the incidence of ROP was 3.2% in the non-supplemented zinc group and 0% in the supplemented group. The incidence of ROP in the Se-supplemented groups was higher, but it did not exceed 33%. In addition to the effects of antioxidant micronutrients, SeP has been shown to inhibit vascular endothelial growth factor-stimulated cell proliferation, tubule formation, and migration in human umbilical vein endothelial cells [41], indicating that SeP may directly affect the induction of neoangiogenesis in ROP and represent a therapeutic target.

Two potential limitations should be kept in mind when analyzing the results of our study. Firstly, our study did not take into account the influence of the genotypes of the studied genes in mothers on the predisposition to the analyzed neonatal diseases. However, the *SELENOP* rs3877899 genotype in the mother can be speculated to influence uteroplacental-fetal interactions, pregnancy complications, and preterm delivery since mothers of children with the rs3877899AA homozygote genotype are also carriers of the rs3877899A allele. Secondly, the presence of polymorphisms in other genes that may influence the occurrence of preterm birth, particularly those involved in the inflammatory process, cannot be excluded. The *SELENOS* rs34713741 SNP may be associated with circulating levels of proinflammatory cytokines, but the regulation of inflammation is complex, so further studies that include more genes involved in the inflammatory process are warranted to fully elucidate the etiology of preterm birth and the related complications. The strong point of the study was the high statistical power of the genetic analysis for ROP. A post hoc analysis of the study revealed a statistical power of 96.3%, 97.6%, and 45% for the association between the *SELENOP* rs3877899AA genotype and the incidence of ROP requiring treatment, ROP treatment failure, and ELGA, respectively. In the GxE interaction study, the estimated statistical power was 96.1% for *SELENOP* rs3877899GA+AA genotypes and 99.9% for ELGA in the analysis of the proliferative type of ROP (requiring treatment). For the analysis of ROP undergoing spontaneous remission, the estimated statistical power was <80% and 94.4%, respectively. The statistical power of the tested interaction effect in both analyses was lower than 80%. These findings confirm the significance of the *SELENOP* rs3877899A allele and ELGA in the pathogenesis of severe forms of ROP. Nonetheless, validation of the presence of GxE interactions between these risk factors requires replication of studies on a larger population of prematurely born children.

In conclusion, this association study supports the importance of selenoproteins and potential Se deficiency in the pathogenesis of ROP. Analysis of *SELENOP* SNPs may be helpful in identifying those preterm infants with ELGA who are at particularly high risk for the progression of ROP to advanced stages and who may benefit from preventive Se supplementation. For a better understanding of the role of selenoproteins in the pathogenesis of ROP, it could be important to investigate the relationship between SeP levels, changes in newborns’ condition after birth, and the timing of ROP occurrence.

## 4. Materials and Methods

### 4.1. Study Population

We carried out a prospective study in which premature infants born between 22 and 32 weeks of gestation were screened for the presence and advancement of ROP. Based on ophthalmic criteria, a population of 190 children was selected, with similar proportions of subjects with no ROP, spontaneously regressing ROP, and ROP requiring treatment. All the infants were hospitalized at the Department of Neonatology in the Gynecology and Obstetrics Clinical Hospital (Level III Hospital) of the Poznan University of Medical Sciences between 2009 and 2017. Homogenic consistency was maintained as all infants were of Caucasian origin. The study design excluded multiple pregnancy births, infants with chromosomal abnormalities or TORCH infections (rubella, herpes, cytomegalovirus, toxoplasmosis, etc.), and infants without antenatal steroid therapy.

### 4.2. Clinical Features and Outcomes

The following clinical factors were analyzed: gestational age (GA, weeks), birth weight (BW, grams), sex, time of preterm rupture of the fetal bladder (days), mode of delivery (vaginal delivery vs. cesarean section), Apgar scores at 1 and 5 min, blood transfusions, surfactant and oxygen therapy, and duration of mechanical ventilation (days). In addition to ROP, the evaluated comorbidities of prematurity included sepsis, jaundice, RDS, IVH, BPD, NEC, and PVL. The employed criteria for diagnosing RDS, NEC, IVH, and BPD have been described in detail previously [42]. The diagnosis of PVL was based on abnormal US or MRI results. Sepsis and neonatal jaundice were diagnosed on the basis of clinical symptoms and standardized laboratory tests. The diagnosis of intrauterine hypotrophy was based on the World Health Organization criteria, namely, fetal weight below the 10th percentile according to percentile grids for the appropriate gestational age. Subgroups of newborns with ELGA and ELBW were distinguished based on a birth age lower than 28 weeks and a body weight of less than 1000 g.

The methods of ROP diagnosis, classification, and treatment have been described in detail by Chmielarz-Czarnocinska et al. [43]. In short, the infants were screened with indirect ophthalmoscopy by a team of three trained pediatric ophthalmologists specializing in ROP. The examination was performed under topical anesthesia with proxymetacaine after pupil dilation with tropicamide 1% and phenylephrine 2.5% drops applied three times. The initial funduscopic examination was performed four weeks after birth, and follow-up examinations were performed every 7–10 days during the hospital stay or every 1–3 weeks, depending on the severity of the lesions, after the patient’s discharge from the hospital (until full retinal vascularization, lesion stabilization, or the need for treatment). Based on the Early Treatment for Retinopathy of Prematurity guidelines, treatment was provided within 72 h of detection of severe ROP using laser photocoagulation (LP) and/or Lucentis therapy (intravitreal ranibizumab injections, IVR). Peripheral retinal ablations were carried out using a 810-nm diode laser with confluent burns (Iris Medical OcuLight SL, IRIDEX Corporation, CA, USA) under general anesthesia. The patients were examined on the day following the procedure and every 7–10 days until total ROP regression was ascertained, changes were stabilized, or retreatment was necessary. All treated infants were monitored by a team of participating physicians.

### 4.3. Genotyping

The biological material for genetic research was obtained from buccal swabs taken from the inner surface of the child’s cheek. DNA isolation was performed using the innuPREP Swab DNA Kit (Analytik Jena, Jena, Germany). The presence of the studied SNPs—*SELENOP* rs3877899 (c.700G>A, p.Ala234Ser), *SELENOP* rs7579 (c.*14G>A, 3’UTR variant), *SELENOS* rs34713741 (2 kb C>T 5’UTR variant), and *GPX4* rs713041 (c.660T>A; p.Leu220=)—was determined with predesigned TaqMan SNP genotyping assays containing probes for allelic discrimination (respectively: C2841533_10, C8806056_10, C3091980_10, and C2561693_20), using the ABI 7900HT Fast Real-Time PCR System (Life Technologies, Carlsbad, CA, USA). The *SELENOS* rs34713741 SNP was selected for this research instead of the most studied variant of the promoter region (−105G>A, rs28665122), as the commercially available predesigned assay gave inconclusive results. Both SNPs are in complete linkage disequilibrium in the Central European population; hence, the tested SNP can be considered a linked marker. The genotyping success rate was 99%. Samples with failed genotype calls at more than 1 SNP were excluded from the analysis.

### 4.4. Statistical Analysis

Genotype frequencies were tested for the Hardy–Weinberg equilibrium (HWE) using a *χ^2^* test available at https://ihg.helmholtz-muenchen.de/cgi-bin/hw/hwa1.pl (accessed on 12 December 2022). The individual effects of the studied factors on the onset and progression of ROP were evaluated using linear regression analysis to determine the significance of the trend. The regression beta coefficient (β) was determined to compare the strength of the influence of the studied factors. In this analysis, the groups were assigned the following designations: 0—no ROP; 1—ROP requiring treatment; and 2—ROP not requiring treatment. Univariate analyses assessing the effect of the studied factors on the treatment outcome used the *χ*^2^ or Fisher’s test for qualitative variables and the *t*-test or Mann–Whitney U test for quantitative variables. Kaplan–Meier curves were used to visualize the relationship between age at birth and the onset and course of ROP. The values of odds ratios (ORs) and 95% confidence intervals (95%CIs) were also calculated for the genotypes. Gene-environment interactions were assessed using the method described by Botto and Khoury [44], which is based on the use of a multivariate table with a two-by-four layout. The assessment was performed using univariate analyses (*χ*^2^ or Fisher’s test). The Rothman synergy index (S), which indicates a deviation from the additive model of interactions, was also determined. The interpretation of the coefficient values is as follows: S = 1 indicates no interaction; S < 1 indicates a relative decrease; and S > 1 indicates an increase in the strength of the interactions between the two factors. A multivariable stepwise regression analysis was performed to evaluate the potential independent effect of the studied variables on the occurrence and progression of ROP. Quanto software (developed by The Division of Biostatistics, Keck School of Medicine of the University of Southern California, Los Angeles, CA, USA) was used for the post hoc statistical power analysis. All other analyses were performed using STATISTICA version 13.0 (Dell, Tulsa, OK, USA) and GraphPad Prism version 6.04 (GraphPad Software, Boston, MA, USA) software. The observed differences were considered significant at *p* < 0.05.

## Figures and Tables

**Figure 1 ijms-24-07570-f001:**
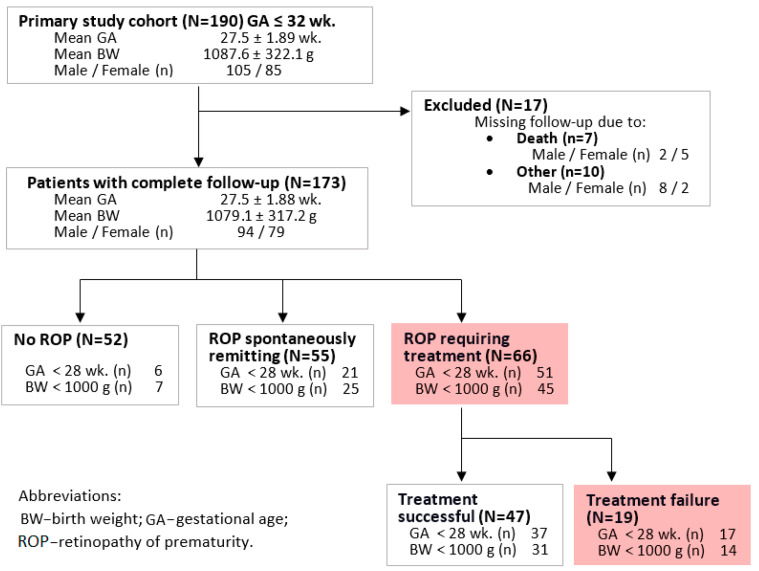
Consolidated scheme of the study’s inclusion criteria and population characteristics.

**Figure 2 ijms-24-07570-f002:**
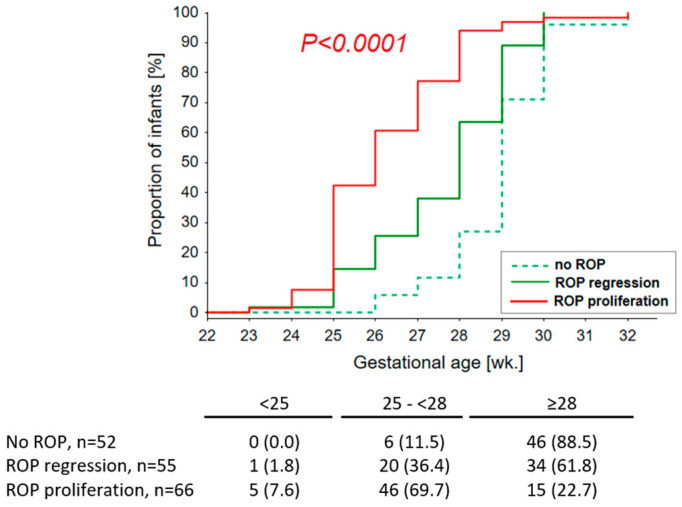
The relationship between birth age and the occurrence and course of ROP.

**Figure 3 ijms-24-07570-f003:**
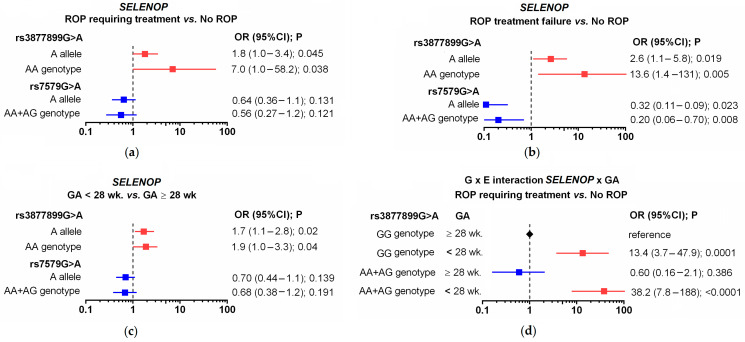
Summarized results of univariate analyses of associations and gene-environment interactions (G × E) between selenoprotein-encoding genes, gestational age below 28 weeks (GA < 28 wk.), and ROP. Association of the *SELENOP* rs3877899 and rs7579 SNPs with the occurrence of ROP requiring treatment (**a**), ROP treatment failure (**b**), and GA < 28 wk. (**c**). Multiplicative G × E interaction between the *SELENOP* rs3877899 genotype and GA < 28 wk. affects the incidence of ROP requiring treatment (**d**).

**Figure 4 ijms-24-07570-f004:**
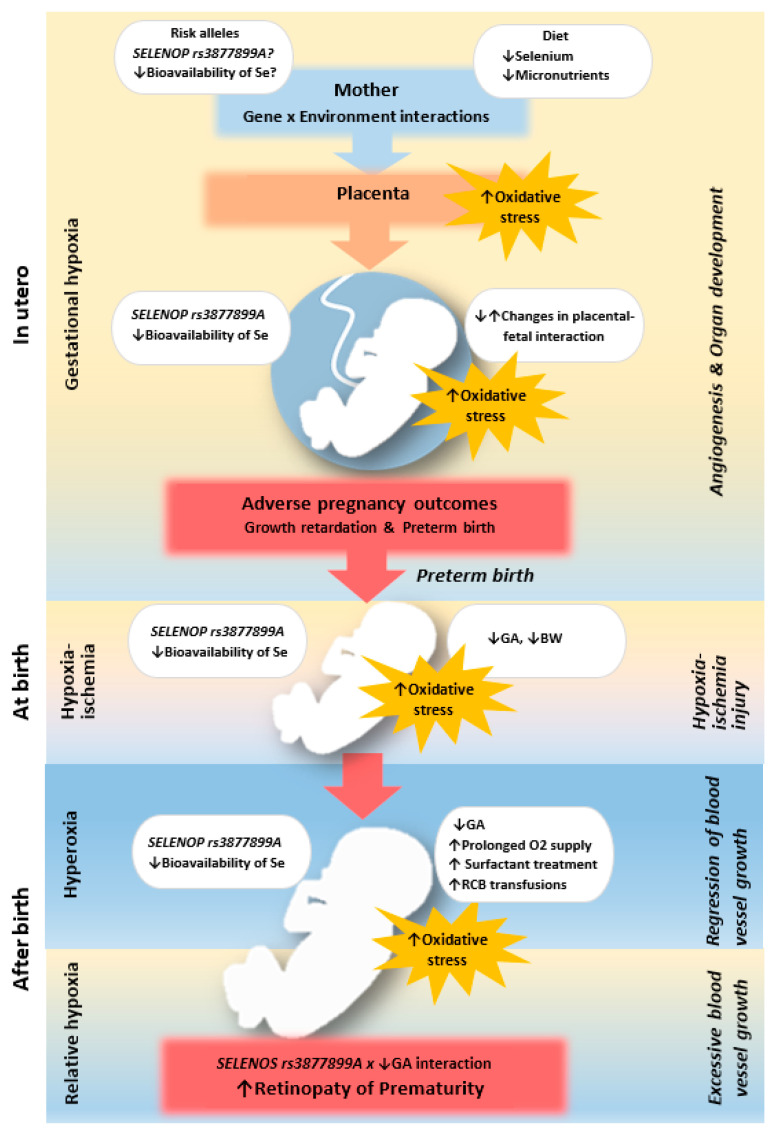
Overview of the postulated mechanism through which a child’s *SELENOP* genotype influences in utero and postnatal homeostatic disorders leading to ROP occurrence. The placenta facilitates important in utero functions, including nutrient transfer, metabolism, gas exchange, neuroendocrine signaling, growth hormone production, and immune control. Insufficient dietary selenium supply, as observed in European countries, may contribute to inadequate protection of the placenta from oxidative stress during pregnancy, affecting fetal development. This may be exacerbated by genetic risk factors present in the mother and child, resulting in reduced Se bioavailability. Among them are variants in the gene encoding selenoprotein P, the body’s main Se store and transporter. Abnormalities in fetal development can lead to changes in placental-fetal interactions and result in premature birth. During and after delivery, the immature newborn is exposed to oxidative stress associated with changes in environmental oxygen concentration. The use of oxygen therapy after birth inhibits angiogenesis in the eye. After the termination of this treatment, relative hypoxia occurs, which is a strong stimulus for vascular growth and the development of ROP. Extreme prematurity and carrying the *SELENOP* rs3877899 A allele is associated with depression of the antioxidant barrier and the occurrence of oxidative damage. Oxidative stress is strongly related to ROP pathogenesis because of the susceptibility of the phospholipid-rich retina to reactive oxygen species that can be generated in high or low oxygen. Other mechanisms of the protective effects of selenoprotein P may include the inhibition of VEGF-stimulated neoangiogenesis.

**Table 1 ijms-24-07570-t001:** Demographic and clinical parameters associated with ROP and failure of its treatment.

Parameter	Incidence and Outcome of ROP				ROP Requiring Treatment		
	I No ROP N = 52	II ROP Not Requiring Treatment N = 55	III ROP Requiring TreatmentN = 66	*β*, *p_for trend_* I/II/III	IIIa ROP Treated Successfully N = 47	IIIb ROP Treated Unsuccessfully N = 19	*p* IIIa vs. IIIb
Gestational age [wk.]							
Mean (SD)	28.9 (1.2)	27.7 (1.7)	26.2 (1.6)	**−0.591, <0.0001**	26.4 (1.8)	25.9 (1.6)	**0.375**
Range	26–32	23–30	23–34		23–32	24–29	
Body mass [g]							
Mean (SD)	1317.7 (326.6)	1044.3 (266.9)	920.2 (226.1)	**−0.509, <0.0001**	943.4 (242.9)	862.6 (170.1)	**0.191**
Range	640–2080	490–1600	500–1700		500–1700	570–1160	
Intrauterine hypotrophy; n (%)	3 (7.7)	7 (12.7)	2 (3.1)	−0.051, 0.504	2 (4.4)	0 (0.0)	**0.358**
Male sex; n (%)	26 (50.0)	31 (56.4)	37 (57.8)	0.063, 0.412	28 (62.2)	9 (47.4)	**0.279**
Apgar 1; Median (Q1; Q3)	6 (4; 8)	4 (2; 6)	3 (3; 6)	**−0.323, <0.0001**	3 (1; 6)	3.5 (1; 6)	**0.769**
Apgar 5; Median (Q1; Q3)	7 (7; 8)	7 (6; 8)	7 (6; 7)	**−0.267, <0.001**	7 (6; 7)	7 (5; 7)	0.960
No. of RBC transfusions; Mean (SD)	1.4 (1.12)	2.8 (2.2)	5.5 (2.8)	**0.554, <0.0001**	4.5 (2.0)	8.0 (3.6)	**0.026**
Risk factors at birth							
Ruptured fetal bladder; n (%)	19 (36.5)	18 (32.7)	22 (34.9)	−0.012, 0.873	14 (31.8)	8 (42.1)	**0.440**
Ruptured fetal bladder period [d]; Mean (SD)	3.6 (6.4)	4.6 (10.9)	5.1 (14.0)	0.058, 0.449	6.7 (16.4)	1.6 (2.6)	**0.192**
Delivery by caesarean section; n (%)	29 (55.8)	27 (49.1)	29 (45.3)	−0.085, 0.269	22 (48.9)	7 (36.8)	**0.384**
Parameters related to respiratory failure							
Surfactant treatment; n (%)	17 (23.7)	27 (41.8)	46 (71.9)	**0.327, 0.0005**	33 (73.3)	13 (68.4)	**0.695**
Resuscitation; n (%)	34 (65.4)	49 (89.1)	62 (96.9)	**0.354, 0.0001**	43 (95.6)	19 (100.0)	**0.358**
Mechanical ventilation; n (%)	32 (61.5)	43 (78.2)	65 (98.5)	**0.388, <0.0001**	46 (97.9)	19 (100.0)	**0.529**
Mechanical ventilation period [d]; Mean (SD)	6.1 (9.0)	18.8 (19.2)	33.7 (20.9)	**0.548, <0.0001**	33.9 (21.5)	33.3 (19.7)	**0.908**
Complications of prematurity; n (%)							
NEC	6 (11.5)	17 (31.5)	24 (37.5)	**0.234, 0.002**	16 (35.6)	8 (42.1)	**0.628**
BPD	7 (13.5)	27 (49.1)	46 (71.9)	**0.477, <0.0001**	34 (75.6)	12 (63.2)	**0.321**
PDA	8 (15.4)	21 (38.2)	21 (32.8)	0.149, 0.052	15 (33.3)	6 (31.6)	**0.893**
IVH	22 (42.3)	40 (72.7)	54 (84.4)	**0.364, 0.0001**	36 (80.0)	18 (94.7)	**0.142**
RDS	23 (44.2)	26 (47.3)	48 (75.0)	**0.262, 0.001**	34 (75.6)	14 (73.7)	**0.877**
DWMI	1 (9.9)	7 (12.7)	8 (12.5)	0.144, 0.061	7 (15.6)	1 (5.3)	**0.262**
Sepsis	6 (11.5)	14 (25.9)	28 (43.7)	**0.296, <0.001**	20 (44.4)	8 (42.1)	**0.866**
Neonatal jaundice	49 (94.2)	47 (85.5)	57 (89.1)	−0.064, 0.332	39 (86.7)	18 (94.7)	**0.353**

Abbreviations and symbols: BPD—bronchopulmonary dysplasia; DWMI—diffuse white matter injury; IVH—intraventricular hemorrhage; NEC—necrotizing enterocolitis; PDA—patent ductus arteriosus; RDS—respiratory distress syndrome; ROP—retinopathy of prematurity.

**Table 2 ijms-24-07570-t002:** The effect of gene-environment interaction between the *SELENOP* genotype and extremely low gestational age (ELGA) on the incidence and severity of retinopathy of prematurity (ROP).

Risk Factors		Incidence and Outcome of ROP			Statistical Analysis OR (95%CI); *p*	
rs3877899	ELGA	I No ROP	II ROP not requiring treatmen	III ROP requiring treatment	II vs. I	III vs. I
GG	≥28 wk.	28 (52.8)	20 (36.4)	11 (16.7)	reference	reference
GG	<28 wk.	4 (7.5)	12 (21.8)	21 (31.8)	4.2 (1.2–14.9); 0.027	13.4 (3.7–47.9); 0.0001
GA+AA	≥28 wk.	18 (34.0)	14 (25.5)	4 (6.1)	1.1 (0.44–2.7); 0.854	0.6 (0.16–2.1); 0.386
GA+AA	<28 wk.	2 (3.8)	9 (16.4)	30 (45.5)	**6.3 (1.2–32.4); 0.028**	**38.2 (7.8–187.7); < 0.0001**
OR expected from additive model (G+E) ^a^_,_ Synergy index ^b^					4.3; 1.6 ↑	12.9; 3.1 ↑↑

a-Expected OR = OR1 Observed (GG and <28 wk.) + OR2 Observed (GA+AA and ≥28 wk.) − 1; b-The Rothman synergy index (S).

**Table 3 ijms-24-07570-t003:** Multivariable analysis—independent predictors of the occurrence and progression of ROP.

Risk Factors	Impact on the Occurrence and Progression of ROP					
	Model 1			Model 2		
	*β*	Influence [%]	*p*	*β*	Influence [%]	*p*
Number of RCB transfusions	0.315	30.7	<0.00001	0.323	30.7	<0.00001
ELGA	0.346	8.5	<0.00001	0.232	8.5	<0.00001
Surfactant treatment	0.152	2.1	0.017	0.169	2.1	0.017
rs3877899GA+AA and ELGA	NA			0.172	1.8	0.024
Regression model summary						
Influence [%]	41.3			43.1		
Significance	F(3.2) = 38.7; *p* < 0.00001			F(4.164) = 31.1; *p* < 0.00001		

## Data Availability

All data generated or analyzed during this study are included in this published article (and its Appendix A).

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
