# Peer review of "SELENOP rs3877899 Variant Affects the Risk of Developing Advanced Stages of Retinopathy of Prematurity (ROP)"

_ijms, 2023, doi:10.3390/ijms24087570_

Round 1
Reviewer 1 Report
Retinopathy of prematurity (ROP) is a leading cause of childhood blindness. This study evaluates the association of selenoprotein polymorphism and the risk of ROP and other comorbidities, and identified the SELENOP rs3877899A allele as a risk factor for ROP development and poor prognostic in extremely preterm infants.
Overall this is a highly clinically relevant piece of work. The manuscript is well written and the high statistical power of the genetic analysis is impressive. I only have the following points that the authors can consider commenting on or discussing in more details.
11. The association between SELENOP rs3877899A allele and the reduced selenium bioavailability has been speculated. I am wondering was circulating plasma selenium concentration examined in infants of SELENOP variants?
22. Postnatal Se supplementation in very low birth weight infants failed to improve neonatal outcome of ROP (Darlow et al., 2003)---any comments?
33. A previous study in UK showed women who carry the SELENOP rs3877899A allele are better able to maintain selenium status during pregnancy (Mao et al.,2015). I am curious how would that affect the Se status and ROP outcome for their fetus?
Author Response
R1. We would like to thank Reviewer for careful reading our manuscript and providing comments that helped us to improve the text.
Q1. The association between SELENOP rs3877899A allele and the reduced selenium bioavailability has been speculated. I am wondering was circulating plasma selenium concentration examined in infants of SELENOP variants?
Replay
Reliable analysis of circulating biomarkers was not possible in the population of preterm infants due to blood transfusions given to many children, especially those in the worst clinical condition, which could affect Se/SeP/Gpx3 activity levels. However, previous studies in adults have confirmed the influence of the studied SNPs on Se bioavailability in the body. This effect is due to the influence of polymorphisms on the synthesis ratio of the two isoforms of SeP, which differ in size and Se content (with 1 vs 10 Sec residues), leading to an impact on Se transport and the production of all other selenoproteins. To improve clarity, we have included an appropriate explanation and citations in the manuscript (lines 264-266)."
We add:
These SNPs were found to affect selenium’s bioavailability for the synthesis of all other selenoproteins by influencing the body’s Se status and transport, as well as the effectiveness of supplementation [28-30].
- Méplan, C.; Crosley, L.K.; Nicol, F.; Beckett, G.J.; Howie, A.F.; Hill, K.E.; Horgan, G.; Mathers, J.C.; Arthur, J.R.; Hesketh, J.E. Genetic polymorphisms in the human selenoprotein P gene determine the response of selenoprotein markers to selenium supplementation in a gender-specific manner (the SELGEN study). Faseb j 2007, 21, 3063-3074, doi:10.1096/fj.07-8166com.
- Mao, J.; Vanderlelie, J.J.; Perkins, A.V.; Redman, C.W.; Ahmadi, K.R.; Rayman, M.P. Genetic polymorphisms that affect selenium status and response to selenium supplementation in United Kingdom pregnant women. Am J Clin Nutr 2016, 103, 100-106, doi:10.3945/ajcn.115.114231.
- Kurokawa, S.; Bellinger, F.P.; Hill, K.E.; Burk, R.F.; Berry, M.J. Isoform-specific binding of selenoprotein P to the β-propeller domain of apolipoprotein E receptor 2 mediates selenium supply. J Biol Chem 2014, 289, 9195-9207, doi:10.1074/jbc.M114.549014.
Q2. Postnatal Se supplementation in very low birth weight infants failed to improve neonatal outcome of ROP (Darlow et al., 2003)---any comments?
Replay
We add information on the results of RCT and comments to text (see below Q3).
Q3. A previous study in UK showed women who carry the SELENOP rs3877899A allele are better able to maintain selenium status during pregnancy (Mao et al.,2015). I am curious how would that affect the Se status and ROP outcome for their fetus?
Replay
In fact, the previous study in the UK showed that women who are carriers of the SELENOP rs3877899A allele are able to better maintain their selenium status during pregnancy, as well as after supplementarion [1]. “Selenium status” means in this case increase in GPx3 activity, so antioxidant capacity. These results are promising in the context of potential “protective” supplementation during pregnancy. However, an RCT needs to be conducted to draw a clear clinical conclusion. The impact of maternal selenium supplementation, according to Cochrane Database of Systematic Reviews, from 20 weeks gestation on the outcome for very preterm infants deserves investigation, although given that very preterm infants constitute only around 1% of births, at least 5,000 of pregnant women would have to be recruited for that study, which is challenging.
Q2 and Q1: Changes in text:
Randomized clinical trials have shown that postnatal Se supplementation in very low birth weight infants can reduce one or more episodes of sepsis. However, such supplementation has not improved neonatal outcomes in terms of ROP, survival, and lung diseases [39]. These outcomes may be due to insufficient dosages of supplementation, lack of maternal supplementation during pregnancy, and modifying effects of the SELENOP genotype. The influence of genotype was previously demonstrated in the UK population, where pregnant women who carry the rs3877899A allele could better maintain their Se status during pregnancy and were more responsive to Se supplementation than carriers of the rs3877899G allele [40]. The impact of the rs3877899A allele on Se supplementation was also observed in women from the general population [28]. These results are promising for the potential use of protective supplementation during pregnancy. However, developing a personalized therapy approach presents a challenge as it would require clinical trials involving several thousand pregnant women. The supplementation of another micronutrient, zinc, in preterm infants was significantly associated with reduced mortality but not comorbidities like BPD, ROP, bacterial sepsis, or NEC [41]. A limitation of the aforementioned studies in the infant cohorts was the low prevalence of the diseases studied. For example, in one study on ROP involving 193 premature infants, the incidence of ROP was 3.2% in the non-supplemented zinc group and 0% in the supplemented group. The incidence of ROP in the Se-supplemented groups was higher, but it did not exceed 33%. In addition to the effects of antioxidant micronutrients,
- Darlow, B.A.; Austin, N.C. Selenium supplementation to prevent short-term morbidity in preterm neonates. Cochrane Database Syst Rev 2003, 2003, Cd003312, doi:10.1002/14651858.Cd003312.
- Mao, J.; Vanderlelie, J.J.; Perkins, A.V.; Redman, C.W.; Ahmadi, K.R.; Rayman, M.P. Genetic polymorphisms that affect selenium status and response to selenium supplementation in United Kingdom pregnant women. Am J Clin Nutr 2016, 103, 100-106, doi:10.3945/ajcn.115.114231
28. Méplan, C.; Crosley, L.K.; Nicol, F.; Beckett, G.J.; Howie, A.F.; Hill, K.E.; Horgan, G.; Mathers, J.C.; Arthur, J.R.; Hesketh, J.E. Genetic polymorphisms in the human selenoprotein P gene determine the response of selenoprotein markers to selenium supplementation in a gender-specific manner (the SELGEN study). Faseb j 2007, 21, 3063-3074, doi:10.1096/fj.07-8166com.
Typos and genotype labeling in Figure 3d have also been corrected (GG genotype instead of AA genotype).
Reviewer 2 Report
*More than 52% of infants born before 28 weeks of gestation carried the rs3877899A 149 allele compared with 38% in infants born after at least 28 weeks of gestation (P=0.02).
This result of the study looks very interesting and should be discussed in the paper. Is this allele a risk factor of pre-term delivery?
**Table 2.
GG <28 wk. 4 (7.5) 12 (21.8) 21 (31.8) 4.2 (1.2 - 14.9); 0.027 13.4 (3.7 - 47.9); 0.0001
GA+AA <28 wk. 2 (3.8) 9 (16.4) 30 (45.5) 6.3 (1.2 - 32.4); 0.028 38.2 (7.8 - 187.7); < 0.0001
Is the difference between the two significant?
***
There is an in build problem in ROP studies. Some of them take the 28 Week as the separating line for evaluation (surfactant level) other use the 30 week (the end of the retinal hyperoxic (relative to intrauterine oxygen levels) phase. It would be interesting to compre whether the change of criteria from 28 W to 30W makes any difference.
Author Response
R2. We would like to thank Reviewer for careful reading our manuscript and providing comments that helped us to improve the text.
Comments and Suggestions for Authors
Q*More than 52% of infants born before 28 weeks of gestation carried the rs3877899A 149 allele compared with 38% in infants born after at least 28 weeks of gestation (P=0.02).
This result of the study looks very interesting and should be discussed in the paper. Is this allele a risk factor of pre-term delivery?
Replay:
The results of our study also relate to the risk of preterm birth, as the reviewer rightly pointed out. We observed the association between rs3877899A allele and ELGA (Figure 3c and Supplemental Table S2). It is very interesting result as confirm significant role of selenoproteins in development and preterm delivery, showing that selenoproteins can be involved in the maintenance of pregnancy through uteroplacental-fetal interactions. For the brevity of the text, we have actually omitted from the discussion which result is discussed, which may lead to inaccuracies. We have added this information in the text.
[Line 267: Our findings regarding the association of the SELENOP rs3877899A allele with ELGA…. ]
We also provide a more detailed discussion on the importance of selenoproteins in development (see publication 12 cited in the Introduction).
The importance of selenoproteins for the development is illustrated by the embryonic lethal phenotype of knockout mice of the genes of selenoprotein biosynthesis pathway. In human, carriers of mutations in these genes (including SECISBP2, TRU-TCA1-1 and SEPSECS) display the severe multisystem phenotypes of global developmental delay with features of myopathy and cerebellar and optic atrophy [12].
Q**Table 2.
GG <28 wk. 4 (7.5) 12 (21.8) 21 (31.8)
GA+AA <28 wk. 2 (3.8) 9 (16.4) 30 (45.5)
Is the difference between the two significant?
Replay
Referring to this data we observed a statistical trend towards a higher frequency of carriers of the A allele in children with ROP, but the difference between the groups was not significant: II vs. I (4, 2, 12, 9): OR = 1.5; P = 1.0; III vs. I (4, 2, 21, 30) OR = 2.9; P = 0.388; II+III vs. I OR = 2.4; P = 0.4. We used the 2 by 4 tables analysis (Botto, L.D. and Khoury, M.J. 2001) to separately measure the effect of the environmental factor, genotype, and joint effect. We chose this model instead of comparing genotypes separately in younger and older children subgroups because it closely matched the observed relationships. The size of subgroups was too small for analysis, considering the strength of the effect (the study not have 80% statistical power to detect significant differences for the tested SNPs).
Q*** There is an in build problem in ROP studies. Some of them take the 28 Week as the separating line for evaluation (surfactant level) other use the 30 week (the end of the retinal hyperoxic (relative to intrauterine oxygen levels) phase. It would be interesting to compare whether the change of criteria from 28 W to 30W makes any difference.
Replay
As we understand well, this question aims at a better characterization of the involvement of selenoproteins in the pathomechanism of ROP development. In the study group, screening for ROP was carried out at intervals taking into account the condition of the retina, at least weekly intervals, which seems to allow for the proposed assessment. However, given that the study is partly retrospective and this time points were not the focus of the study, we cannot provide a credible analysis of this topic. The collected data, on the other hand, allow us to assess the time of onset of the disease in the study groups, which we supplement in results and add literature data (Discussion) regarding average age at diagnosis. Our population falls within the range of standard observations.
Results: [ In the studied group, ROP was diagnosed on average at 52.9 ± 12.4 days of life (7.6 weeks), at a mean post-menstrual age (PMA) of 34.4 ± 2.0 weeks].
Discussion: [It is diagnosed on average at 8 wk. of life (34-35 PMA).]
In addition, in such a study, it seems that instead of the genotype, the SeP protein level should be tested in 28W and 30W . We consequently add the suggestion
Conclusion: [For a better understanding of the role of selenoproteins in the pathogenesis of ROP, it could be important to investigate the relationship between SeP levels, changes in newborns' condition after birth, and the timing of ROP occurrence.]
If this question concerns more the cut-off point in the analysis of interactions between genotype and GA (28 vs 30 weeks), then the analysis performed for 30 weeks does not indicate a multiplicative interaction effect, but rather an additive effect, consequently the use of ELGA (<28 weeks), is more suitable for the studied data.
Typos and genotype labeling in Figure 3d have also been corrected (GG genotype instead of AA genotype).